# Physiological and Ankle Functions Are Discriminating Factors for the Risk of Falls in Women in Treatment of Osteoporosis

**DOI:** 10.3390/ijerph191912643

**Published:** 2022-10-03

**Authors:** Renata Gonçalves Pinheiro Correa, Anna Raquel Silveira Gomes, Victoria Zeghbi Cochenski Borba

**Affiliations:** 1Internal Medicine, Federal University of Paraná, Curitiba 80060-000, Brazil; 2Prevention and Rehabilitation in Physical Therapy Department, Master’s and PhD Program in Physical Education, Federal University of Paraná, Curitiba 80060-000, Brazil; 3Internal Medicine Department and Master’s and PhD Program, Endocrine Division (SEMPR), Federal University of Paraná, Curitiba 80060-000, Brazil

**Keywords:** elderly, osteoporosis, intrinsic factors, falls, fractures, treatment

## Abstract

Introduction: Elderly women with osteoporosis are at risk of falls and fractures. Objective: To compare the intrinsic factors of falls, including ankle evaluation, in a group of elderly women in treatment for osteoporosis compared with a control group. Methods: A cross-sectional study of elderly women in treatment for osteoporosis (TG) was paired with a control group (CG) not in treatment. All groups completed a questionnaire and underwent a bone mineral density test; the mini-mental state examination (MMSE); physical performance tests; lower-limb strength and power, ankle, and muscle architecture evaluations; and a physiological profile assessment (PPA). Results: A total of 128 women were included (68 TG, 60 CG); the mean age was 71.55 ± 3.07 years; TG had a worse performance in the intrinsic factors in the MMSE, plantarflexions range of motion, gait speed, plantarflexions peak isometric strength, and short physical performance battery (*p* < 0.05 for all). PPA stratification (proprioception and lower-limb strength) presented a greater risk of falls in the TG, with proprioception increasing the risk by 2.4 times. Conclusion: Patients undergoing treatment for osteoporosis are influenced by intrinsic factors of falls, many being present in the CG. PPA and ankle strength and flexibility tests are more discriminative for evaluating fall risks in patients in treatment for osteoporosis.

## 1. Introduction

Aging is associated with muscle and bone loss, which reduces sensorimotor skills and physiological capabilities, such as vision, muscle strength, proprioception, and balance, increasing the risk of falls and fractures, especially in women with osteoporosis. The ankle strategy is the first balance attitude to prevent falls and fractures. Little is known about the changes in ankle musculoskeletal functions associated with physiological capabilities in postmenopausal women with osteoporosis. The ankle dorsiflexor and plantiflexors are the first line of defense in postural control while standing; it was shown that during reactive pace, the ankle muscles are activated before the knee and hip muscles [1]. Aging deteriorates the biomechanical function of the ankle for different reasons, such as a loss of muscle strength and mobility, modifying the anatomy of the ankle and feet, which can increase the risk of falls and fractures [2].

This information contributes to the planning of successful intervention strategies. Elderly women have a high prevalence of osteoporosis (30%), which increases with age, and when associated with decreased functional capacity and balance, causes a higher incidence of falls and fractures. Falls cause 87% of all fractures in elderly patients with osteoporosis, usually due to low-impact injuries [3,4]. It is estimated that 80% of menopausal women suffer from fractures because they do not receive adequate treatment [5], and those undergoing medical treatment often have a low adherence, which may be related to treatment issues [6]. Other key factors for fractures are muscle strength, power, overall functional stability, coordination, and performance, all associated with an increased risk of falls and fractures [7]. Guidelines from geriatric societies recommend annual screening for the risk of falls in people aged 65 years and older [8].

In 2010, the prevalence of osteoporosis and low bone mass in non-institutionalized individuals aged 50 years and older was estimated at 10.2 and 43.4 million, respectively. The prevalence is higher in women (15.4%) compared to men (4.3%) and increases with age from 5.1% at the age of 50 years to 26.2% at age ≥ 80 years [9,10].

A systematic review that aimed to provide evidence for the primary prevention of falls in postmenopausal women concluded that the association of sarcopenia and osteoporosis increases the risk of low-energy fractures [11]. Osteoporosis does not present specific clinical manifestations until the first fracture, and therefore a detailed physical examination should be performed to identify the predictive factors for future fractures. There is a scarcity of studies that analyze the intrinsic factors present in patients under treatment for osteoporosis based on the frequency of falls. We showed elsewhere that patients in the community under treatment for osteoporosis had most intrinsic factors for falls altered, and the musculoskeletal function of the ankle could differentiate frequent-fall patients from non-frequent fallers.

Although anti-resorptive therapy is effective for elderly women with osteoporosis at low and moderate fracture risks, a systematic review with meta-analysis showed that current screening strategies used for fracture prevention based on bone mineral density (BMD), FRAX, or International Osteoporosis Foundation criteria recommend interventions for only 5 to 25% of the participants, with the need to incorporate a more robust screening method involving the musculoskeletal system to propose a more assertive intervention [12].

Thus, the present study aims to compare the intrinsic factors of falls, including ankle evaluation, in a group of elderly women in treatment for osteoporosis and to compare these with a control group.

## 2. Materials and Methods

### 2.1. Study Design

This is a cross-sectional study using women from the community in treatment for osteoporosis. It was developed based on the Strengthening the Reporting of Observational Studies in Epidemiology (STROBE) [13] and approved by the research ethics committee of the Hospital de Clínicas da Universidade Federal do Paraná with the protocol number 3320592, and conducted from January 2019 to December 2021, in agreement with the Declaration of Helsinki.

### 2.2. Participants

Women (≥65 years) in treatment for osteoporosis were recruited during their routine visit to the outpatient clinic of the Endocrinology and Metabology Service of the Hospital de Clínicas of the Federal University of Paraná (SEMPR) and composed the treatment group (TG). Women who matched by age and race; who never treated for osteoporosis; and who were recruited among friends, relatives of patients, and from the community composed the control group (CG). The exclusion criteria were chronic renal failure, functional dependence, incapacity to perform the requested tests, patients with chronic diseases under current treatment that could interfere with the assessment of the variables, and medications affecting bone mass other than treatment for osteoporosis.

The sample size was calculated using the G*Power 3.1.9^®^ program (National Institutes of Health, Bethesda, MD, USA). Considering the sample (*n* = 68) for the treatment (*n* = 60) and control groups, a size effect of 0.5 (mean effect), and type-I error (5% error rate), the sampling power (1-β) of the study was 0.99.

### 2.3. Study Procedures

The evaluations were divided into two days. On the first day, after signing the informed consent form following the rules of the Declaration of Helsinki of 1975, the included individuals answered a structured questionnaire developed for the study to collect demographics, disease characteristics, treatment, comorbidities, lifestyle habits, amount of dietary calcium intake from dairy foods, self-reported history of falls, and low-energy trauma fractures. Calcium and vitamin D supplementation were evaluated as present or absent, and the daily calcium intake was classified based on dietary recommendations. Missing information was collected from patients’ medical charts. The mini-mental state examination (MMSE) test was applied [14], followed by functional and isometric strength tests. On the second day, muscle architecture was evaluated, and a dual-energy X-ray absorptiometry (DXA) was performed. All the assessments were performed by the same researcher. The intraclass correlation coefficient (ICC) was performed to test the intra-rater reliability as well as the standard error measurement (SEM). Both were conducted in a pilot study with a sample of 15 participants for each test. The final values demonstrated the average of three measurements, with 0 indicating no reliability and 1 indicating perfect reliability.

### 2.4. Fall History

Fall was defined as any event resulting from a body change that caused the individual to inadvertently fall to the ground. This definition did not encompass the result of a violent blow, sudden paralysis, loss of consciousness, or epileptic seizure [15]. The evaluation of the fall history in the past 12 months occurred through a self-report method. Individuals were also asked about the location of, reasons for, and consequences of the fall. Participants were classified as non-fallers (0 falls), fallers (1 fall), and recurrent fallers (≥2 falls) based on the number of falls they experienced in the previous year.

### 2.5. Sample Characterization

To characterize the sample, the following variables were investigated: age, education, marital status, occupation, income, ethnicity, medication in use, treatment time, comorbidities, fractures, visual acuity, hearing acuity, use of orthotics, metallic prostheses, and history of conservative and/or surgical treatments.

### 2.6. Anthropometric Assessment and Cognitive Screening

Weight was measured using a calibrated anthropometric mechanical scale (Plenna^®^) with a capacity of 150 kg, with the participant barefoot and wearing underwear. Height was measured with a vertical stadiometer (Tonelli Gomes^®^), with a graduation of 1 mm and a maximum height of 2.20 m. The body mass index (BMI) was calculated by dividing the participant’s weight (in kg) by their height squared (in meters). The BMI results were classified as underweight (BMI < 18.5 kg/m^2^), normal (>18.5 kg/m^2^ < BMI < 24.9 kg/m^2^), pre-obese (25 kg/m^2^ < BMI < 29.9 kg/m^2^), and obese (BMI > 30 kg/m^2^) [16].

### 2.7. Bone Mineral Density

A bone mineral density (BMD) test was performed by dual-energy X-ray absorptiometry (DXA) in a Horizon A machine (serial number 201383, Hologic, Bedford, MA, USA) in the lumbar spine (L1-L4), total hip, femoral neck, and total body. The BMD results were expressed as g/cm^2^ and evaluated according to the recommendation of the International Society for Clinical Densitometry (ISCD), and they were classified as normal or low (osteopenia or osteoporosis). The trabecular bone score (TBS) was obtained from all DXA lumbar scans (L1-L4, not excluded in the BMD measurement) using TBS iNsight software version 3.0.2.0 (MediMaps, Geneva, Switzerland), as well as versions 2.2 and 3.0. It was stratified according to the studies conducted on the male population in Latin America. The microarchitecture was considered as degraded when values were ≤1.230, partially degraded when values were >1.230 and <1.310, and normal when they were ≥1.310 [17]. All DXA evaluations were performed by a certified operator and analyzed by a certified radiologist. We calculated the participants’ appendicular lean mass (ALM) as the sum of the lower and upper limbs’ lean mass obtained from the total body evaluation, and the relative skeletal muscle index (RSMI) as the ratio of ALM to ASM over the squared-height (ASM/heiht^2^) cutoff points for women < 6.0 kg/m^2^ [18].

### 2.8. Cognitive Screening

Cognition was assessed with the MMSE. The cutoff scores used to detect cognitive disorders were 18/19 for uneducated individuals and 24/25 for educated individuals, considering seven dimensions: temporal orientation (5 points), spatial orientation (5 points), immediate memory (3 points), attention and calculus (5 points), delayed memory recall (3 points), language (8 points), and visual-constructive ability (1 point) [14].

### 2.9. Physical Performance

The functional mobility and the risk of falls were evaluated by the time to get up and go (TUG) test, which included rising from a chair without the help of the arms and walking at a safe pace for 3 m. A physical therapist guided the participants with the following verbal command: stay seated, and at the command “go”, stand up from the chair and walk at a comfortable and safe pace until you reach that cone 3 m away. Turn around at that point, return to your chair, and sit down once more. It was scored following the cutoff points for an age range of 60–69 years: 8.1 (7.1–9.0) s; 70–79 years old: 9.2 (8.2–10.2) s; 11.3 (10.0–12.7) s [18], ICC 0.983 and SEM 0.63.

The short physical performance battery (SPPB) is a set of measures that combines the results of gait speed, standing up from a chair, and balance tests [19]. The scores ranged from 0 (worst performance) to 12 (best performance).

The sit-to-stand test (SST) is a repeated chair lift in which the subject stands up and sits down as quickly as possible five times without stopping. The cutoff values for falls were established according to the participants’ age as follows: 70–79 years, >12.6 s; ICC 0.348 and SEM 7.35 [19].

The balance test involved standing without support for 10 s with the feet together next to each other for about 10 s, and then proceeding to semi-tandem and tandem positions. The gait speed in meters per second (m/s) was used to calculate the gait speed recorded in SPPB; it was considered as low when ≤0.8 m/s when walking at the usual pace between the two cones [19].

### 2.10. Strength

A handgrip dynamometer (Saehan SH5001^®^) was used to measure the hand-grip strength (HGS). The participant was seated on a chair without an armrest, with their feet on the ground and hips and knees at a 90° flexion, with shoulders positioned in adduction and neutral rotation, elbows at a 90° flexion, with the forearm and wrist in a neutral position. The participant was instructed to perform the maximum hand-grip movement of the dominant upper limb for 3 s, performing three maximum movements with 1–2 min of rest between each repetition. The highest result obtained from three attempts was recorded (in kilogram force [kgf]). The dynamometer’s grip was individually adjusted according to the size of the participant’s hands. A result less than 16 kg indicated low muscle strength [19]. The ICC and SEM were 0.961 and 0.83, respectively.

### 2.11. Ankle Musculoskeletal Function

Ankle range of motion (ROM) during plantar flexion and dorsiflexion movements performed actively and without previous heating was evaluated with a goniometer (Carci^®^) positioned with the axis below the lateral malleolus, with the fixed arm aligned with the participant’s fibular head and the movable arm parallel to the lateral edge of the foot. An average of three repetitions of each joint movement was performed, and the ROM was expressed in degrees. The cutoff values of 26 ± 6.3° for dorsiflexion (ICC and SEM of 0.935 and 5.16, respectively) and 57 ± 7.2° for plantarflexion (ICC and SEM of 0.938 and 1.80, respectively) were indicative of decreased mobility [20]. 

The peak isometric strength (PIS) of ankle dorsiflexion and plantiflexors was assessed using a handheld dynamometer (Lafayette Hand-Held Dynamometer, model 0115, Lafayette Industry^®^ (System 4 Pro™, United Kingdom Lafayette Instrument Company, USA) programmed to measure the peak isometric strength for 5 s with sound signals Available online: (http://www.lafayetteevaluation.com/products/01165a-hand-held-dynamometer-kit accessed on 12 August 2022). After a patient’s choice of the dominant limb, two repetitions with the non-dominant limb were performed for test training, followed by three repetitions with the dominant limb with an interval of 30 s. The cutoff points for peak dorsiflexion strength were: 60–69 years old, 27.82 kg; 70–79 years old, 25.10 kg (ICC and SEM of 0.627 and 1.71, respectively), and 27.15 kg for plantiflexors (ICC and SEM of 0.797 and 2.43, respectively [21].

### 2.12. Muscle Architecture

Muscle architecture was assessed by an ultrasound (B-mode ultrasound, with a 3.8 mm linear-array probe, 11 MHZ) of the gastrocnemius muscle, measuring 40% of the popliteal line and medial malleolus. Three parameters of the muscular architecture of the medial gastrocnemius muscle were analyzed: the muscle thickness (MT) (ICC 0.998 and SEM 0.006), calculated considering the distance between the superficial and deep aponeuroses; the pennation angle (PA) (ICC 0.973 and SEM 0.03), as the angle between the muscular fascicle and the deep aponeurosis; and the fascicle length (FL) (ICC 0.923 and SEM 0.07) as the length of the fascicle trajectory between the fascicle insertions in the superficial and deep aponeuroses. The largest and best fasciculus was used and, when it was longer than the surface of the probe, the fascicle line was extrapolated and calculated using the formula proposed in the literature [22,23].

Images were analyzed using Image J software (National Institutes of Health, Bethesda, MD, USA; Available online at: https://imagej.nih.gov/ij/ accessed on 4 August 2022). Cutoff points for muscle thickness were: 1.80 ± 1.12, fascicle length: 4.07 ± 2.98, and pennation angle: 25.40 ± 18.30 [23,24].

### 2.13. Falls Risk Assessment

The fall risk was evaluated with the physiological profile assessment (PPA short form), a series of simple tests for vision, proprioception, muscle force, knee extension, reaction time, and postural sway. A score higher than 0 indicated an increased risk of falling, 0–1 indicated a mild increased risk, 1–2 indicated a moderate increased risk, 2–3 indicated an increased risk, and >3 indicated a very marked increased risk. Visual contrast sensitivity was measured using the Melbourne edge test; proprioception was measured using a lower-limb matching task, where errors in degrees were recorded using a protractor inscribed on a vertical, clear, acrylic sheet placed between the legs; quadricep strength in kilograms was assessed isometrically in the dominant leg while participants were seated with their hips and knees at 90 degrees; simple hand-reaction time in milliseconds was measured using a light stimulus and finger pressure as a response; and postural sway (path length in millimeters) was measured using a sway meter recording displacements of the body at the level of the pelvis while participants stood on a foam rubber mat with their eyes open [25]. The ICC and SEM were 0.967 and 0.32, respectively.

### 2.14. Statistical Analysis

Data normality was assessed using the Kolmogorov–Smirnov and Shapiro–Wilk tests. Data were presented as mean ± standard deviation (SD) or median (minimum and maximum) values, or as absolute and relative frequencies, as appropriate. Categorical variables were described by frequency and percentage. Quantitative variables were compared using the Student’s *t*-test for independent samples or with the nonparametric Mann–Whitney test, with an effect-size cutoff point of negligible effect [>0.20–<0.20]; small effect [>0.21–<0.39]; medium effect [>0.40 and <0.79]; and high effect [0.80]. Categorical variables were compared using Fisher’s exact or the chi-squared tests. The assumption of homogeneity of variance was evaluated by Levene’s test. The bootstrapping procedures were performed (1000 resampling; 95% CI BCa) to obtain a higher reliability of the results and correct possible deviations in the normality, distribution sizes of the groups, and to present a 95% confidence interval. A multivariate Poisson model analysis was performed to identify the main intrinsic factors related to falls. All the analyses were performed using the Statistical Package for the Social Sciences SPSS^®^ v.14.1 (SPSS, Inc, Chicago, IL, USA).

## 3. Results

During the study period, 785 patients were seen in the outpatient clinic, 340 were invited to participate, 192 declined, 39 did not meet the inclusion or exclusion criteria, and 41 did not perform the study tests. Of the remaining patients, 68 were included in TG. Furthermore, 60 women recruited from the community and not in treatment for osteoporosis were included in the CG (Figure 1). Most participants were white (87.5%) with a mean age of 71.55 ± 3.07 years. The median time of treatment for osteoporosis was 2.34 years (2.04–2.73), and bisphosphonates were the only medication used in the TG. 

### 3.1. Comparison between Groups

The groups were similar in age and race, but the TG was heavier than the CG, had more comorbidities, and consumed more medication. The number of falls and past fractures were similar between the groups. Nevertheless, the BMD and TBS were lower in the TG and a higher number of patients were classified as having osteoporosis by the BMD, whereas in the CG, most patients had osteopenia; *p* < 0.05 for all (Table 1). The complete sociodemographic data are depicted in Table 1.

### 3.2. Intrinsic Factors for Falls

The physical-performance test results were variable, with a lower performance rate in the TG in GS (*p* < 0.004) and SPPB tests (*p* < 0.000). The CG took more time to perform the TUG test compared to the TG (*p* < 0.008). The HGS and SST were similar between the groups (Table 2).

Both groups had an impairment of ankle musculoskeletal function. The ankle ROM of dorsiflexion and plantarflexion was low in both groups, with a lower range in plantarflexion in the CG compared to the TG; *p* < 0.000. Additionally, the ankle PIS strength decreased in both groups and was even lower in the TG (*p* < 0.019 vs. CG). 

The three parameters of muscle architecture measured by ultrasound were equally reduced in both groups (Table 2).

The physiological risk of falls evaluated by the PPA showed a low or very low risk of falls in both groups, but a greater proportion of moderate/high risk of falls in the treatment group. This difference was mainly observed in the component’s proprioception and strength, which was more affected in the TG (Table 3).

The multivariate analysis (Poisson regression) considered falls as the dependent variable and all significant intrinsic factors as independent variables (TUG, GS, SPPB, plantarflexion ROM, plantarflexion PIS, proprioception, and knee extensor strength) and observed that proprioception increased the chance of falls by 2.4% (95% CI 1.006–1.042); IRR: 1.024, *p*: 0.010.

## 4. Discussion

When considering the ample physical–functional evaluation not previously conducted on this type of patients, this study reported a worsened performance of intrinsic factors with a higher risk of falls in patients undergoing treatment for osteoporosis, compared to the control group.

Important aspects of fall risks, such as a history of falls and fractures, strength, ankle mobility, and muscle architecture, were similar between patients with osteoporosis under treatment and those not treated. One explanation was the similarity of clinical characteristics between the groups, including race and age, variables related to falls and fractures. The exception was BMI, which was lower in the CG and known to be associated with worse muscle and bone mass [26]. Indeed, 100% of patients from both groups had strength, ankle mobility, and muscle architecture below the normal cut point, which agreed with the similar number of falls observed among the groups. Although the vast majority of individuals were classified as non-fallers, the literature presents 30% of falls in this age range [26]. The low number of falls observed could mean a bias in the fall-capture method. 

Despite the similar number of past fractures between the groups, the TG had worse BMD and bone quality evaluated by TBS, with a higher number of patients classified as having osteoporosis, which could be compatible with a more ill group of individuals already in treatment for osteoporosis, with a higher number of comorbidities and twice the number of medications.

The altered intrinsic factors in the TG were related to the physical performance (GS, SPPB), balance (SPPB), plantarflexion ROM, and PIS. The previous literature has shown a low physical performance in postmenopausal women with osteoporosis as evaluated by SPPB and low physical fitness in the TUG test, with a greater risk of falls compared to healthy elderly women [27,28].

The low ankle function of TG showed a deficiency in the first approach to overcome the imbalance, confirmed by the poor performance and worse balance in this group. This strategy was not previously investigated in patients with osteoporosis, but it was described in community-dwelling older women [29]. Other studies on the elderly achieved the same results. One study observed a loss in ROM with a significant negative correlation between ankle and metatarsophalangeal joint flexibility [30], and the other described a significant association between balance and reduced ankle inversion/eversion movement [31]. These results corroborate the changes in ankle function and the relationship with balance disorders and risk of falls in elderly women. Ankle range of motion during gait has also been associated with falls [31]. In our study, participants who fell more than twice in the previous year had a lower ankle ROM compared to non-fallers. A different result was observed in a prospective observational study conducted by Menz et al., which did not observe a relationship between ankle ROM and falls, though the fallers exhibited reduced ankle flexibility [31,32].

PPA, a method used to evaluate physiological functions (sight, strength, proprioception, reaction time, and postural sway), exhibited worse proprioception and lower-limb strength in the TG, which classified the patients as having a high risk of falls. PPA was a critical evaluation used in this study; it showed that proprioception doubled the risk of falling as already presented in another study conducted on elderly women in the community [33]. It could be a new tool to evaluate fall risks for women in treatment for osteoporosis and a new strategy used for fall prevention in these patients [33].

Other studies presented observations concerning elderly women in the community using a battery of tests, such as PPA or PPA itself [34]. They related the impairments with age [35] and observed a reduction in the muscle strength of knee extensors, less dependence on the visual field, and greater body sway [25]. Another study conducted in Australia concluded that a marked deficit in any system may be sufficient to predispose elderly women to falls, and that the combination of mild or moderate deficiencies in several physiological domains may also increase the risk of falls associated with certain comorbidities, such as labyrinthitis, cataracts, diabetic neuropathy, and sarcopenia [36].

The limitations of this study included the cross-sectional design, the possibility of memory bias when providing the number of medications, fracture and fall history, and a possible bias in the selection of the patients and controls. The positive aspects of the current study were the sample size in both groups, the ample evaluation, screening a high number of risk factors for falls, using low-cost instruments, and the focus on women with osteoporosis. 

In conclusion, this study showed that patients undergoing treatment for osteoporosis were influenced by the intrinsic factors of falls, many of which were present in the women of the community but not in the treatment. The physiological factors for falls and tests of ankle strength and flexibility were more discriminative in evaluating the fall risks for patients in treatment for osteoporosis, and they should potentially be incorporated into future assessments of these patients to direct more assertive preventive interventions. 

## Figures and Tables

**Figure 1 ijerph-19-12643-f001:**
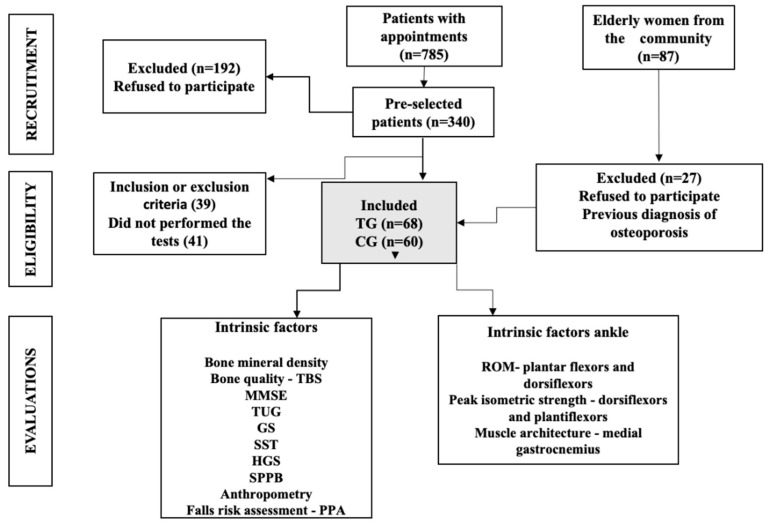
Flow diagram of the selection of the study participants. TG, treatment group; CG, control group; TBS, trabecular bone score; MMSE, mini-mental state examination; TUG, timed up and go test; GS, gait speed; SST, sit-to-stand test; HGS, hand-grip strength; SPPB, short physical profile battery; PPA, physiological profile assessment; ROM, range of motion.

**Table 1 ijerph-19-12643-t001:** Anthropometric data of patients and controls.

Variables	TG (*n* = 68) Mean ± SD	CG (*n* = 60) Mean ± SD	*p*
Age (years)	71.98 ± 3.47	71.13 ± 2.41	0.113
Race			
White	88.23% (*n* = 60)	98.33% (*n* = 59)	0.577
Black	11.70% (*n* = 8)	1.66% (*n* = 1)	0.600
Height (m)	1.54 ± 0.06	1.58 ± 0.08	0.002
Weight (Kg)	64.13 ± 11.13	68.85 ± 10.54	0.015
BMI (kg/m^2^)	26.52 ± 3.71	21.77 ± 3.35	<0.000
Underweight	-	16.70% (*n* = 10)	<0.000
Normal	41.17% (*n* = 28)	56.70% (*n* = 34)	<0.000
Pre-obese	38.23% (*n* = 26)	23.40% (*n* = 14)	<0.000
Obese	20.59% (*n* = 14)	-	<0.000
Comorbidities (N)	1.46 ± 1.30	0.76 ± 0.96	<0.000
Hypertension	46.37% (*n* = 32)	53.33% (*n* = 32)	0.246
Hyperglycemia	79.71% (*n* = 55)	76.66% (*n* = 46)	0.855
Dyslipidemia	60.86% (*n* = 42)	43.33% (*n* = 26)	0.085
DM2	79.71% (*n* = 55)	76.65% (*n* = 46)	0.855
Medications (N)	4.10 ± 2.96	2.30 ± 2.56	0.000
Osteoporosis treatment			
Alendronate	50.0% (*n* = 34)	-	
Risedronate	2.94% (*n* = 2)	-	
Pamidronate	23.53% (*n* = 16)	-	
Zoledronic acid	2.94 % (*n* = 2)	-	
No treatment ^#^	20.59% (*n* = 14)	-	
Treatment time (years)	1.94 ± 4.38	-	
BMD (g/m^2^)			
Spine	0.768 ± 0.13	0.901 ± 0.19	<0.000
Femoral neck	0.647 ± 0.90	0.715 ± 0.27	0.057
Total hip	0.748 ± 0.11	0.693 ± 0.15	0.023
BMD classification			
Normal	2.90% (*n* = 2)	21.70% (*n* = 13)	<0.000
Osteopenia	20.30% (*n* = 14)	58.30% (*n* = 35)	<0.000
Osteoporosis	76.80% (*n* = 57)	20.00% (*n* = 12)	<0.000
Low	97.10% (*n* = 67)	78.30% (*n* = 47)	<0.000
TBS score	1.23 ± 0.10	1.30 ± 0.14	0.001
Normal	30.00% (*n* = 14)	56.60% (*n* = 34)	<0.000
Partially degraded	38.60% (*n* = 27)	23.30% (*n* = 14)	<0.000
Degraded	41.40% (*n* = 29)	20.00% (*n* = 12)	<0.000
Past fracture % (N)	15.30% (*n* = 11)	13.50% (*n* = 8)	0.455
1	13.40% (*n* = 9)	11.40% (*n* = 7)	0.142
≥2	1.10% (*n* = 1)	0.90% (*n* = 1)	0.142
Falls classification			
Non-faller	78.5% (*n* = 53)	81% (*n* = 49)	0.866
Faller	16.5% (*n* = 11)	11%(*n* = 7)	0.866
Recurrent faller	5% (*n* = 4)	6% (*n* = 4)	0.866
MMSE	26.50 ± 3.11	27.96 ± 1.62	0.001

Abbreviations: values expressed as mean ± standard deviation, relative (%) and absolute (numbers) frequencies; TG: treatment group; CG: control group; BMI: body mass index; DM2: type-2 diabetes melitus; BMD: bone mineral density; TBS: trabecular bone scores; MMSE: mini-mental state examination; ^#^ no treatment at the moment.

**Table 2 ijerph-19-12643-t002:** Intrinsic factors for falls in the treatment and control groups.

Variables	TG (*n* = 69)		CG(*n* = 60)		*p*
	Mean ± SD	Cohen D	Mean ± SD	Cohen D	
	*n* (%)		*n* (%)		
**Physical performance**					
TUG (s)	10.06 ± 2.70	2.2	11.43 ± 3.04	0.48	0.008
Reduced (<9.2 s)	61.4% (*n* = 43)		75.9% (*n* = 44)		0.090
GS (s)	0.90 ± 0.17	4.11	0.81 ± 0.14	0.58	0.004
Low (≤0.8 m/s)	74.3% (*n* = 52)		46.6% (*n* = 27)		0.001
SPPB (score)	9.30 ± 2.40	4.87	7.88 ± 1.36	0.72	0.000
0 (worst performance)					
12 (best performance)					
**Strength**					
SST (s)	12.30 ± 2.74	0.3	12.79 ± 3.10	0.17	0.343
Low (%)	45.7% (*n* = 32)		41.4% (*n* = 53)		0.182
HGS (kgf)	21.24 ± 3.78	4.62	22.46 ± 4.24	0.31	0.087
Low (≤18 kgf)	11.4% (*n* = 8)		8.6% (*n* = 11)		0.174
**Ankle ROM (°)**		0.06		0	
Dorsiflexors	22.16 ± 6.10		21.98 ± 4.09	0.8	0.989
Low (<26 ± 6.3°)	100%		100%		1
Plantiflexors	46.46 ± 7.17	4.13	41.23 ± 5.86	0.55	0.001
Low (<57 ± 7.2°)	100%		100%		1
**Muscle architecture**					
Medial gastrocnemius					
MT	1.40 ± 0.23	11.23	1.37 ± 0.26	0.12	0.452
FL	4.43 ± 0.88	4.61	4.36 ±0.97	0.08	0.686
PA	19.09 ± 4.70	4.24	20.45 ± 4.36	0.3	0.092
PIS—ankle					
Dorsiflexion	7.78 ± 2.96	3.47	8.23 ± 1.21	0.2	0.275
Plantarflexion	10.41 ± 4.19	0.69	11.87± 2.45	0.42	0.019

Values expressed as mean ± standard deviation, relative (%) and absolute (numbers) frequencies; TG, treatment group; CG, control group; TUG, timed up and go test; s, seconds; TUG normal values by age: 60–69 years: 8.1 s (s), 70–79 years: 9.2 s, ≥80 years or older: 11.3 s; GS, gait speed; m, meters; SPPB, short physical performance battery; SST, sit-to-stand test; SST normal values by age: 60–69 years: 11.4 s, 70–79 years: 12.6 s, 80–89 years: 12.7 s; HGS, hand-grip strength; kgf, kilogram force; ROM, range of motion, MT, muscle thickness; FL, fascicle length; PA, pennation angle; MG, medial gastrocnemius; PIS, peak isometric strength.

**Table 3 ijerph-19-12643-t003:** Physiological profile assessment stratification of falls.

	TG (*n* = 69) Mean ± SD *n* (%)	D-Cohen	CG (*n* = 60) Mean ± SD *n* (%)	*p*	Cohen-D
Visual contrast	16.92 ± 2.15	2.22	17.21 ± 1.57	0.391	0.15
Excellent/good	8.6% (*n* = 6)		1.6% (*n* = 1)	0.047	
Fair/bad	91.30% (*n* = 63)		93.3% (*n* = 56)	0.046	
Proprioception	2.51 ± 1.54	9.53	1.80 ± 0.90	0.002	0.56
Good/fair	84.05% (*n* = 58)		95.00% (*n* = 57)	0.018	
Bad	11.94% (*n* = 11)		5.00% (*n* = 3)	0.018	
Strength LM	23.82 ± 5.82	5.16	27.60 ± 6.17	0.001	0.67
Excellent/good	69.56% (*n* = 48)		93.33% (*n* = 56)	0.002	
Fair/bad	30.43% (*n* = 21)		6.66% (*n* = 4)	0.003	
Reaction time	391.04 ± 111.59	4.48	394.86 ± 109.40	1	0.03
Excellent/good	17.39%(*n* = 12)		18.33% (*n* = 11)	1	
Fair/bad	82.60%(*n* = 57)		81.66% (*n* = 49)	1	
Postural sway	322.60 ± 147.99	0.55	319.33 ± 157.60	0.564	0.02
Excellent/good	100.00% (*n* = 69)		98.33% (*n* = 59)	0.166	
Falls risk	2.22 ± 1.17	2.98	1.56 ± 0.94	0.050	0.62
Very low	5.7% (*n* = 4)		26.66% (*n* = 16)	0.003	
Low	73.91% (*n* = 51)		65.00% (*n* = 39)	0.002	
Moderate/very high	14.49% (*n* = 10)		5.00% (*n* = 3)	0.002	

Values expressed as mean ± standard deviation, relative (%) and absolute (numbers) frequencies; TG, treatment group; CG, control group; LM, lower members.

## Data Availability

The data presented in this study are available on request from the corresponding author. The data are not publicly available due to ethical reasons and future use.

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
