# Peer review of "Physiological and Ankle Functions Are Discriminating Factors for the Risk of Falls in Women in Treatment of Osteoporosis"

_ijerph, 2022, doi:10.3390/ijerph191912643_

Round 1

Reviewer 1 Report

The manuscript aims to compare intrinsic factors of falls, including ankle evaluation, in a group of elderly women in treatment for osteoporosis and compare them with a control group.

As a strong point of the study, I would highlight the sample size in both groups (TG n=68; CG n=60), as well as the study itself focusing on women with osteoporosis.

From my point of view the article is relevant and correct and I would not have any comments to make as a proposal for modification or improvement. Therefore, I convey my congratulations on the work to the authors.

Author Response

RESPONSE TO REVIEWER 1

The manuscript aims to compare intrinsic factors of falls, including ankle evaluation, in a group of elderly women in treatment for osteoporosis and compare them with a control group.

Point 1: As a strong point of the study, I would highlight the sample size in both groups (TG n=68; CG n=60), as well as the study itself focusing on women with osteoporosis.

From my point of view the article is relevant and correct and I would not have any comments to make as a proposal for modification or improvement. Therefore, I convey my congratulations on the work to the authors.

Answer 1: Thanks for alerting us, we add the information as suggested in the discussion, as "The positive aspects of the study were the sample size in both groups, the ample evaluation, screening a great number of risk factors for falls and using low-cost instruments and the focus on women with osteoporosis."

Reviewer 2 Report

The aim of the study is to analyse and compare the intrinsic factors of falls among women undergoing treatment for osteoporosis and women without treatment.

Among the strengths of the study are the number of variables analysed and the interest of the subject of the study due to the hospitalisation costs it generates.

Among the weaknesses that can be attributed to the study are the sample selection criteria for both the treatment group and the control group (it is likely that there is an important bias in the selection of the sample).

Here are some comments to consider before publication:

Line 17: where it says 129 women it should say 128 (68 TC, 60 CG).

Line 209: reference physiological profile assessment (PPA-short-209 form) (21)

Line 241: The numbers do not match. Where it says 69 patients it should say 68.

Line 244: add time units (years).

Table 1: Totals do not match BMI, osteoporosis treatment, past fracture in treatment group and control group.

Literature 21 and 33 are the same (reference correctly).

Literature 23 and 27 are the same (reference correctly).

All the literature needs to be reviewed.

Author Response

RESPONSE TO REVIEWER 2

Point 1: Line 17: where it says 129 women it should say 128 (68 TC, 60 CG).

Response 1: Sorry, it was a mistake, we corrected. The number 128 is the correct.

Point 2: Line 209: reference physiological profile assessment (PPA-short-209 form) (21)

Response 2: Sorry if we did not understand the issue with the reference [21]. The reference of the long and short versions is the same. In the article, Lord et al explain the indication for each version, and we decided to choose the short version because it seemed to suit better to an osteoporosis clinic, so the reference [21] is correct.

Point 3: Line 241: The numbers do not match. Where it says 69 patients it should say 68

Response 3: Sorry, we corrected and change to 68.

Point 4: Line 244: add time units (years)

Response 3: Thanks, we added the unit "years " in line 244.

Pont 5: Table 1: Totals do not match BMI, osteoporosis treatment, past fracture in treatment group and control group.

Response 5: You are correct, the numbers and percentages were reviewed and now they are correct, we also check the statistics to be sure they are OK, sorry for the inconvenience.

Pont 6: Literature 21 and 33 are the same (reference correctly).

Response 6: Sorry, you are right, we deleted the reference 33 and re-ordered all references.

Pont 7: Literature 23 and 27 are the same (reference correctly)

Response 7: Sorry, you are right, we deleted the reference 27 and re-ordered all references.

Reviewer 3 Report

The Introduction section should be expanded.

Line 137 - cognition assessment should be under a different subheadings, not bone mineral density

Line 150 - should be rephrased in order to give a meaning to ICC and SEM.

Have the patients met the criteria for sarcopenia diagnostic? 

Author Response

RESPONSE TO REVIEWER 3

Pont 1: The Introduction section should be expanded

Response 1: Following your suggestion we add 2 paragraphs to expand the introduction explaining the importance of ankle and physical evaluation for fractures, as "The ankle dorsiflexor and plantiflexors are the first line of defense in postural control while standing, it was shown that during reactive pace the ankle muscles are activated before knee and hip muscles. [1] Aging decline the biomechanical function of the ankle by different reasons, loss of muscle strength and mobility, modifying the ankle and feet anatomy which can increase the risk of falls and fractures. [2]"

"It is estimated that 80% of menopausal women suffer a fracture because they do not receive adequate treatment [5], and those undergoing medical treatment often have a low adherence, which may be related to treatment issues [6]. Other key factors for fracture are muscle strength, power, overall functional stability, coordination, and performance, all associated with an increased risk of falls and fractures [7]. Guidelines from geriatrics societies recommend annual screening for the risk of falls in people aged 65 years and over [8]".

Point 2: Line 137 - cognition assessment should be under a different subheadings, not bone mineral density

Response 2: We add a new subsection for cognitive assessment named "2.8. Cognitive screening", the sequential numbering of the subsections was adjusted.

Point 3: Line 150 - should be rephrased in order to give a meaning to ICC and SEM

Response 3: We re-phrase the sentence for better understanding as "The intraclass correlation coefficient (ICC) was performed to test the intra-rater reliability as well as the standard error measurement (SEM). Both done in a pilot study with a sample of 15 participants for each test. The final values demonstrate the average of three measurements, with 0 indicating no reliability among raters and 1 indicating perfect reliability among raters"

Point 4: Have the patients met the criteria for sarcopenia diagnostic?

Response 4: Thank you for asking that. Sorry, as this issue was not the objective of this paper, we did not have the results yet. We are evaluating the data and we hope it could be be the topic of a next paper.